# Neuroreceptor Inhibition by Clozapine Triggers Mitohormesis and Metabolic Reprogramming in Human Blood Cells

**DOI:** 10.3390/cells13090762

**Published:** 2024-04-29

**Authors:** Karin Fehsel, Marie-Luise Bouvier, Loredana Capobianco, Paola Lunetti, Bianca Klein, Marko Oldiges, Marc Majora, Stefan Löffler

**Affiliations:** 1Department of Psychiatry and Psychotherapy, Medical Faculty, Heinrich-Heine-University, Bergische Landstrasse 2, 40629 Duesseldorf, Germany; bouvier@arcor.de; 2Department of Biological and Environmental Sciences and Technologies, University of Salento, 73100 Lecce, Italy; loredana.capobianco@unisalento.it (L.C.); paola.lunetti@unisalento.it (P.L.); 3Institute of Bio- and Geosciences, IBG-1: Biotechnology, Forschungszentrum Jülich, Leo-Brandt-Straße, 52428 Jülich, Germany; b.klein@fz-juelich.de (B.K.); m.oldiges@fz-juelich.de (M.O.); 4Leibniz Research Institute for Environmental Medicine (IUF), Auf’m Hennekamp 50, 40225 Düsseldorf, Germany; marc.majora@iuf-duesseldorf.de; 5Clinic for Psychiatry, Psychotherapy and Psychosomatics, Sana Klinikum Offenbach, Teaching Hospital of Goethe University, Starkenburgring 66, 63069 Offenbach, Germany; loehma@t-online.de

**Keywords:** clozapine, apoptosis, oxygen consumption, ATF4, citrate carrier, lipid droplets, HL60 cells, leukocytes

## Abstract

The antipsychotic drug clozapine demonstrates superior efficacy in treatment-resistant schizophrenia, but its intracellular mode of action is not completely understood. Here, we analysed the effects of clozapine (2.5–20 µM) on metabolic fluxes, cell respiration, and intracellular ATP in human HL60 cells. Some results were confirmed in leukocytes of clozapine-treated patients. Neuroreceptor inhibition under clozapine reduced Akt activation with decreased glucose uptake, thereby inducing ER stress and the unfolded protein response (UPR). Metabolic profiling by liquid-chromatography/mass-spectrometry revealed downregulation of glycolysis and the pentose phosphate pathway, thereby saving glucose to keep the electron transport chain working. Mitochondrial respiration was dampened by upregulation of the F0F1-ATPase inhibitory factor 1 (IF1) leading to 30–40% lower oxygen consumption in HL60 cells. Blocking IF1 expression by cotreatment with epigallocatechin-3-gallate (EGCG) increased apoptosis of HL60 cells. Upregulation of the mitochondrial citrate carrier shifted excess citrate to the cytosol for use in lipogenesis and for storage as triacylglycerol in lipid droplets (LDs). Accordingly, clozapine-treated HL60 cells and leukocytes from clozapine-treated patients contain more LDs than untreated cells. Since mitochondrial disturbances are described in the pathophysiology of schizophrenia, clozapine-induced mitohormesis is an excellent way to escape energy deficits and improve cell survival.

## 1. Introduction

Psychiatric disorders affect 970 million people worldwide, representing a significant source of disability. Although the pathophysiology of schizophrenia is still poorly understood, mitochondrial deficits with increased radical production [1,2,3,4] or reduced glucose uptake [5] through disturbed PI3K/Akt signalling [6,7] seem to be part of the underlying mechanisms. Studies on postmortem brains from patients with schizophrenia revealed a dysregulated expression of gene-encoding enzymes of the mitochondrial electron transport chain [8], and aberrant activity of the respiratory chain was shown in the platelets of schizophrenic patients [9]. Increased complex I activity was associated with psychotic symptomology, while its decrease was observed in patients with residual schizophrenia. Therefore, a slowdown of mitochondrial activity and energy consumption by medication should have beneficial effects for patients with schizophrenia. Recently, Hardy et al. [10] showed that the antipsychotic drug (APD) aripiprazole and its metabolites are respiratory chain complex I inhibitors that induce mitochondrial toxicity and decline in cellular ATP and viability. Complex I inhibition might even cause extrapyramidal side effects in 5 to 15% of patients treated with aripiprazole. Other APDs are known to reduce cellular glucose uptake [11,12], which might explain the rapid onset of hyperglycaemia under olanzapine, clozapine, or chlorpromazine treatment [13,14,15].

Standard-of-care treatment of therapy-resistant schizophrenia remains dependent upon antipsychotic medication with clozapine, which demonstrates the best clinical responses via cellular mechanisms that are still unknown. Even addiction, multiple sclerosis [16], Parkinson`s disease [17] and signs of Alzheimer’s dementia [18] were successfully treated with clozapine. Here we studied the metabolic effects of clozapine in order to unravel the underlying mechanisms. In contrast to the dopamine D2 receptor antagonists, clozapine is a pharmacologically ‘dirty’ drug with a unique binding affinity to more than serotonergic, dopaminergic, muscarinic, adrenergic and histaminergic receptor subtypes [19,20]. The therapeutic effectiveness of clozapine may even rely on this broad receptor binding profile. Antagonism of most of these receptors disturbs PI3K/Akt signalling, which guarantees sufficient glucose import into the cells [7]. Glucose uptake is a fine-tuned process that depends on the abundance of glucose transporters (Gluts) on the cell surface. Gluts are stored in vesicles beneath the plasma membrane, and pI3K/Akt signalling causes these vesicles to fuse with the membrane, resulting in the translocation of Gluts into the cell surface. Therefore, reduced AKT kinase signalling results in intracellular glucose deprivation. While olanzapine feeding just tended to decrease Akt phosphorylation, feeding clozapine to rats significantly inhibited Akt signalling [21]. Similarly, receptor antagonism by clozapine markedly reduced in vivo brain glucose metabolism in several brain areas, especially in the cortex [22]. Subsequent intracellular glucose deprivation evokes ER stress [23] as shown for olanzapine and clozapine in hepatocytes [24]. Clinical signs of ER stress are transiently elevated plasma levels of hepatic enzymes [25] and CRP [26] and the febrile inflammatory response [27] at the beginning of clozapine treatment. In this period of about 8 weeks, body metabolism is adapted to the stressor. Energy homeostasis is a delicate balance between energy demand, supply, and storage. Clozapine decreases energy consumption in patients by reducing energy expenditure [28] and body temperature [29]. Excessive energy is stored in fat cells leading to rapid weight gain at the beginning of clozapine treatment [30,31].

Some years ago, unsuspected roles for peripheral neurotransmitters in regulating key physiological functions outside the brain were identified. Serotonin and dopamine have protective effects on the liver and hematopoiesis [32], regulate insulin secretion in the pancreas [33,34], and significantly enhance the proliferation and differentiation of hematopoietic stem cells [35]. Here, we employed the human promyeloid cell line HL60 as an in vitro model for clozapine-induced neutropenia. We studied the multiple metabolic effects induced by low levels of clozapine on the bone-marrow-like HL60 cells (2.5–20 µM). HL60 cells express functional serotonin receptors [36] and still have the potential to differentiate into monocytes as well as granulocytes [37]. Steidl et al. [38] observed overlapping molecular phenotypes of hematopoietic and neuropoietic cells with several receptors for neuromediators on hematopoietic progenitor cells. The cells use glycolysis as well as oxidative phosphorylation for ATP production [39]. In some experiments, we also investigated the effects of cotreatment with the green-tea ingredient EGCG on HL60 cells. This polyphenol has protective effects on cellular and especially mitochondrial functions [40,41]. It displays powerful antioxidative properties and might reduce clozapine-induced superoxide radical production [42].

## 2. Materials and Methods

### 2.1. Cell Culture

Promyeloid HL60 cells (ATCC-CCL-240) were grown in an RPMI1640 medium supplemented with 10% FCS, and 10 mM HEPES (pH 7.4) in a humidified atmosphere of 95% air and 5% CO2 at 37 °C. Cells were seeded at 5 × 10^4^ cells/mL in 12-well plates (Nunc, Wiesbaden, Germany) and treated with clozapine (2.5–20 µM) or vehicle (≤0.2% DMSO) for 8, 24, or 48 h, respectively. In the cotreatment experiments, cells were preincubated with EGCG (Sigma-Aldrich, Taufkirchen, Germany) or 2-deoxyglucose (Sigma-Aldrich) for 30 min before clozapine was added. The clozapine was a generous gift from Novartis Pharmaceuticals (Nuremberg, Germany). It was dissolved in dimethyl sulfoxide (DMSO) at a concentration of 200 mM and then diluted further with ethanol and medium to create a stock solution of 20 mM.

### 2.2. Isolation of Peripheral Blood Mononuclear Cells (PBMCs) and Granulocytes

Venous blood samples (4 mL) were taken from 10 patients under long-term clozapine monotherapy and from 10 healthy controls after having obtained written informed consent from all enrolled subjects. The protocol was carried out in accordance with the Declaration of Helsinki and approved by the ethical committee of our institution. Efforts were made to obtain the samples at the time of medically required blood work to minimise the number of venipunctures. PBMCs and granulocytes were purified by a Ficoll-Paque-Plus-gradient centrifugation after lysis of contaminating red blood cells by a NH_4_Cl lysis solution.

### 2.3. Glucose Assay

HL60 cells were treated with 5 µM and 20 µM clozapine for 24 and 48 h. After centrifugation of the cells, 5 µL of the supernatant were analysed with the GAGO glucose assay (Sigma Aldrich) as recommended by the manufacturer. After 8 h of incubation with 2.5 µM and 5 µM clozapine and/or 10 µM EGCG, the cells were harvested, washed two times with PBS, and lysed in a RIPA-buffer (50 mM Tris-HCl pH 8.0, 150 mM NaCl, 1% Triton-X100, 0.5% sodium deoxycholate, 0.1% SDS). Five µL aliquots of the protein extract were used for the glucose assay.

### 2.4. Protein Concentration

Protein concentrations of cell lysates were determined using a Pierce™ BCA protein assay kit (Thermo Scientific, Waltham, MA, USA) with bovine serum albumin (BSA) standards.

### 2.5. Western Blots

Cells were harvested, washed with PBS, and lysed in RIPA buffer. The lysates were stored at −80 °C. For Western blot analysis; 30 µg protein were separated on 10% Bis-Tris or 16% Tricine gels (Thermo Fisher, Darmstadt, Germany) and transferred to 0.45 µm pore size PVDF membranes (Thermo Fisher). The membranes were routinely stained with Ponceau-S before immunostaining to confirm sample loading and the transfer of equivalent amounts of protein. After blocking unspecific binding by 10% Rotiblock (Roth, Karlsruhe, Germany) in TBS-T for 4 h at 4 °C, the membranes were probed with one of the following 1:500 dilutions of antibodies against activating transcription factor (ATF) 4, Glut1, Glut3, Glut5, phospho-eIF2α, ATPaseIF1 (all from Santa Cruz Biotechnology, Heidelberg, Germany), glycogen synthase kinase (GSK) 3ß, phosphorylated GSK 3ß (New England Biolabs, Frankfurt, Germany) and against the mitochondrial citrate carrier (CIC) [43]. Bands of Akt were detected with the Akt isoform antibody sampler kit and the phospho-Akt thr308 antibody from Cell Signalling (Leiden, The Netherlands). To identify immunoreactive bands, membranes were subsequently incubated with HRP-conjugated secondary antibodies (diluted 1:10,000, Santa Cruz Biotechnology). The immunoreacted proteins were detected by enhanced chemiluminescence, and the amount of proteins was estimated by laser densitometry. Only Glut1 presented as a double band around 55 kD, which was previously observed with different Glut1 antibodies [44]. Membranes were reused several times after stripping with Roti^®^-Free stripping buffer 2.0 for 1 h at room temperature (Carl Roth, Karlsruhe, Germany).

### 2.6. Apoptosis

Morphological changes of the cells were observed under the fluorescence microscope (Axioplan Zeiss, Göttingen, Germany) by Hoechst 33342 staining (8 µg/mL) (Sigma-Aldrich). Apoptotic cells were identified by nuclear condensation, nuclear fragmentation, and apoptotic bodies.

### 2.7. Oil Red Staining

HL60 cells or granulocytes from 10 patients and 10 healthy controls were spotted on glass slides and dried under a cold air stream. Cells were fixed with cold formaldehyde (10%) for 5 h at 4 °C. Oil Red working solution (0.6% Oil Red (Sigma-Aldrich) in 60% isopropanol) was added and incubated for 2 h at room temperature. After extensive washing in PBS, coverslips were fixed on the stained cells with AquaTex (Merck, Darmstadt, Germany). Oil Red-stained lipid droplets were detected by light microscopy and the percentage of cells with LDs was determined.

### 2.8. RT-PCR Analysis

Gene expression was studied by RT-PCR as previously described [42]. The sequences of all primers are listed in Appendix A. After amplification, PCR products were subjected to electrophoresis on 1.8% agarose gels. Bands were visualised by ethidium bromide staining under UV light. Densitometric analysis of the amplification products was performed using the GeneGenius software version 3.02 (Syngene Cambridge, UK).

### 2.9. LC–MS/MS Measurement

All experiments were carried out on an Agilent 1100 series binary HPLC system (Agilent Technologies, Waldbronn, Germany) coupled with an API 4000™ triple quadrupole mass spectrometer (Applied Biosystems/MDS Sciex, Concord, ON, Canada) equipped with a TurboIon spray source. Measurements were performed as described previously [45] with the following minor changes: curtain gas (25) and auxiliary gas temperature (500 °C), and the dwell time for each MS/MS transition was 80 ms. The curtain, nebulizing, auxiliary, and collision gas was nitrogen generated from pressurised air in an NM20ZA nitrogen generator (Peak Scientific, Bedford, MA, USA).

### 2.10. High-Resolution Respirometry

The oxygen consumption of HL60 cells (2 × 10^5^ cells/mL) treated either with or without 20 µM clozapine for 8, 24, and 48 h was recorded in the Oxygraph-2k high-resolution respirometer (Oroboros Instruments, Innsbruck, Austria). All respiration studies were performed in RPMI1640, with 10% FCS at 37 °C and a stirrer at 750 rpm. Following equilibration for at least 20 min with stirring, the chambers were closed, and O_2_ flow was recorded for 30 min. The oxygen consumption was calculated using DatLab software version 7.4 (Oroboros Instruments, Innsbruck, Austria).

### 2.11. Immunohistochemistry

HL60 cells either treated with 20 µM clozapine or 0.2% DMSO were spotted on glass slides and dried. Cells were fixed in acetone for 10 min. After blocking the background with 0.5% BSA for 1 h, the cells were incubated with the hsp27 antibody or CiC antibody (1:500 dilution) in 0.2% BSA at 4 °C overnight, washed three times for 5 min with PBS, and incubated either with Cy3-labeled anti-mouse antibody for 2 h (Thermo-Fisher, Dreieich, Germany) or with the ABC staining kit (Santa Cruz, Heidelberg, Germany) as described by the manufacturer. After extensive washing in PBS, cells were mounted with cover clips and analysed under the microscope.

### 2.12. Determination of VEGF Levels

VEGF levels were determined using commercially available ELISA test kits (Calbiochem, Bad Soden, Germany). All procedures were performed according to the manufacturer’s instructions. The colorimetric VEGF-ELISA covered an assay range from 13.2 to 2000 pg/mL.

### 2.13. Acridine Orange Staining

Acridine orange was used to stain the autophagic vacuoles. Briefly, the cells were stained with acridine orange (4 mg/mL in PBS, Sigma-Aldrich) at room temperature for 1 h, washed twice with PBS, and then observed with fluorescent microscopy [46]. More than 500 stained and unstained cells were counted per incubation medium.

### 2.14. Statistical Analysis

All experiments were done at least in triplicate. The results are expressed as the mean ± standard deviation. A statistical analysis was performed using SPSS22 for Windows. The nonparametric Kruskal–Wallis test was carried out to determine the clozapine effects. Differences with *p* < 0.05 were considered statistically significant.

## 3. Results

### 3.1. Glucose Uptake

HL60 cells were treated with 2.5 and 5 µM clozapine with and without 20 µM EGCG for 8 h. The glucose concentration was determined in the cell lysates. Treatment with 5 µM clozapine significantly reduced the intracellular glucose concentration (Figure 1A). Treatment with EGCG had no additional effect on the glucose content when coincubated with clozapine. For longer incubation times (24 h, 48 h), the glucose concentration was determined in the supernatants of the cell cultures (Figure 1B). After 24 h, the medium contained significantly more glucose, while no difference was found after 48 h. Thus, the glucose metabolism of the HL60 cells was normalised after 2 days.

### 3.2. Apoptosis

Clozapine-induced apoptosis was concentration-dependent. While 5 µM of clozapine promoted cell survival, 50 µM or preincubation with 2-DG or EGCG increased apoptosis in HL60 cells (Figure 1C). Hexokinase activity and further glucose metabolism are inhibited by 2-DG. Although control cells showed a notable increase in apoptosis under 2-DG and EGCG, clozapine-treated cells were much more sensitive to apoptosis under cotreatment.

### 3.3. ER Stress

Reduced Akt activation under clozapine was proven by reduced glucose uptake (Figure 1A,B) as well as decreased phosphorylation of Akt at threonine 308 and decreased GSK3ß phosphorylation [47] (Figure 2A,B; Appendix A). Glucose deprivation under clozapine hampers the correct glycosylation of nascent proteins in the ER, thereby inducing ER stress. Upon ER stress, the unfolded stress response (UPR) stopped general protein synthesis by the phosphorylation of the translation initiation factor eIF2α (Figure 2B) and induced upregulation of specific stress proteins, which manage the different branches of the UPR. Glucose deprivation activates the PERK-ATF4 branch [23,48]. We see 3-fold higher ATF4 levels already under 5 µM clozapine (Figure 2B, Appendix A). After 24 h of clozapine treatment, the glucose transporters Glut3, Glut5 and IF1 (Figure 2A, Appendix A)) were upregulated 1.5-, 1.25- and 1.5-fold, respectively. Expression of hsp27 (Figure 2D) increased as well about 10-fold. Akt expression did not change under clozapine treatment. Protein levels of Akt1 and the insulin-activated Akt2 were detected. The neuron-specific Akt3 was not expressed in HL60 cells (Appendix A).

Vascular endothelial growth factor (VEGF) is induced under ER stress by ATF4 as well as by hypoxia-induced factor (HIF) 1α [48]. Therefore, increased VEGF levels were measured after 48 h of clozapine treatment, while VEGF production was reduced after 24 h (Figure 2E). This slight reduction can be explained by diminished global protein synthesis under ER stress. Phosphorylation of eIF2α creates conditions that favour the synthesis of ATF4 but reduces general translation in ER-stressed, glucose-deprived cells [49]. Although VEGF mRNA levels peak at 24 h, its glycosylation and secretion are delayed. As seen in Figure 2E, VEGF levels in the supernatants of untreated control cells were higher after 48 h compared to 24 h, because HL60 cells constantly produce VEGF [50]. Furthermore, the PERK/ATF4 pathway of the UPR is known to trigger autophagy [51], thereby increasing cell survival. Autophagosomes can be detected by using fluorescent microscopy after incubation with acridine orange. Neither 5 µM clozapine nor 10 µM EGCG led to massive staining by acridine orange (Figure 2F), in contrast to the higher clozapine concentrations of 20 and 50 µM. Similarly, coincubation with the autophagy inhibitor 3-mehyladenine (2 mM) did not increase apoptosis in clozapine-treated HL60 cells (5/20 µM).

### 3.4. Gene-Expression Analysis

Semiquantitative PCR analysis revealed the expression of several stress and metabolically relevant mRNAs after treatment of HL60 cells with 5 and 20 µM of clozapine for 24 h (Figure 2C). Clozapine induced expression of Hif1α and IF1 and lower upregulation of Sirt3, PDK3, and Mn-SOD expression. mRNA levels of GRP78, catalase, and Sirt6 were unchanged. Simultaneous treatment with EGCG solely inhibited the expression of IF1. The housekeeping gene ß-actin was coamplified as a reference.

### 3.5. Metabolic Changes under Clozapine Treatment

HL60 cells were treated with 5 and 20 µM of clozapine for 8 and 24 h. Metabolites of the TCA, PPP, and glycolysis, as well as ATP levels, were determined by LC-MS. In Figure 3, relevant metabolites with considerable changes under clozapine are presented.

After 8 h of incubation (20 µM clozapine) acetylCoA and α-ketoglutarate levels were significantly decreased, while citrate/isocitrate levels were significantly elevated in clozapine-treated cells. This points to a transient inhibition of the enzyme isocitrate dehydrogenase that catalyses the oxidative decarboxylation of isocitrate, producing α-ketoglutarate and CO_2_. All TCA-metabolite levels normalised within 24 h of treatment, but under 20 µM clozapine, it was slower. Vice versa, glycolytic metabolites and metabolites of the PPP were unaffected or even slightly increased after 8 h but were reduced after 24 h of treatment. The ATP pool and the energy charge of the cells decreased within the first 8 h (n.s.) but were replenished after 24 h. The succinate level was slightly increased after 24 h of clozapine treatment. Unfortunately, some results of the metabolic profiling did not reach significance due to high standard deviation, probably due to differences in glucose uptake between cell-cycle phases [52].

Accumulated citrate is transported to the cytoplasm by the mitochondrial CIC. Western blot analysis as well as immunohistochemistry revealed increased levels of CIC in PBMCs of clozapine-treated patients (Figure 4A) and clozapine-treated HL60 cells (Figure 4B).

### 3.6. Lipid Droplets (LD)

LD formation was analysed by Oil Red staining in clozapine-treated HL60 cells as well as granulocytes from 10 patients under clozapine medication and 10 healthy controls (Figure 4C). While 0.9% of untreated cells had weakly stained droplets (A,C), 26% of HL60 cells under 20 µM clozapine showed LDs (B). In blood smears of clozapine-treated patients, 10% of cells were stained. In smears from healthy controls, only 1.7% of the cells showed lipid droplets (D). After in vitro culture with clozapine for 8 h, even 71% of blood cells were stained (E). A graphical presentation of the results is shown in Appendix A. Interestingly, we observed lipid droplets only in neutrophils, not in lymphocytes (Appendix A).

### 3.7. Oxygen Consumption

According to the observed metabolic changes under clozapine, we next ascertained that downregulation of the electron transport chain (ETC) was linked to reduced oxygen consumption. By high-resolution respirometry, we found that oxygen consumption in clozapine-treated HL60 cells was slowly reduced to about 60% (Figure 5A). After 48 h, downregulation of the ETC was complete, and the reduction was stable even after wash-out of clozapine for a further 3 days. Figure 5B shows examples of the comparative measurements.

## 4. Discussion

Through a combination of metabolic and transcriptional profiling, we performed a comprehensive assessment of the effects of clozapine on energy metabolism. Receptor binding by clozapine disrupts the PI3K/Akt pathway, leading to reduced glucose uptake. Similarly, Cochran et al. [53] and Rocha et al. [22] observed reduced local cerebral glucose utilization in clozapine-treated rats. Here, we revealed decreased phosphorylation of threonine 308 of Akt. Furthermore, the significantly reduced intracellular glucose concentration and dephosphorylated GSK3ß point to the missing Akt activation in clozapine-treated HL60 cells. GSK3ß is active when dephosphorylated. It inhibits glycogen synthesis to keep the remaining intracellular glucose available for cell metabolism. The green-tea extract EGCG is known to reduce insulin/Glut4-mediated glucose uptake [54]. But, EGCG had only minor additional effects on clozapine-induced glucose deprivation at a concentration of 10 µM, although HL60 cells express the insulin receptor [55]. However, clozapine itself partially inhibits the insulin-induced glucose uptake [56]. Despite reduced glucose uptake under clozapine, the number of apoptotic cells decreased, indicating that the clozapine-induced stress response is a physiological non-apoptotic response. Vargas et al. [57] confirmed this anti-apoptotic effect of clozapine in the same concentration range (2–20 µM) on spontaneous neutrophil apoptosis. This strongly resembles the health benefits of caloric restriction [58]. However, further disturbance of glucose homeostasis by high concentrations of clozapine (50 µM), 2-DG, or EGCG significantly increased the apoptosis of HL60 cells; 2-DG is a key enzyme inhibitor of hexokinase and interferes with intracellular glucose utilization. Reducing both—glucose uptake and utilization—makes the cells more sensitive to apoptosis under cotreatment. EGCG protects mitochondrial functions and cotreatment and reduces clozapine-linked superoxide radical production (Appendix A). Increased apoptotic rates under clozapine/EGCG cotreatment indicate that the low oxidative stress under clozapine—also seen in neutrophils from clozapine-treated patients [42]—is essential for cell survival. Our data are in good agreement with the results of Contreras-Shannon et al. [59]. They observed mitochondrial dysfunction, ATP decline, and increased apoptosis in several cell lines under clozapine concentrations ≥ 50 µM.

Moderate glucose deprivation induces ER stress [23], which in turn, activates survival pathways in HL60 cells that reduce cell death under 5–20 µM clozapine. ER stress is counteracted by activation of the unfolded protein response (UPR). In the early stage of the UPR, the ER resident kinase PERK phosphorylates the translation factor eIF2α. Phosphorylated eIF2α then selectively activates ATF4 translation, while general protein synthesis is transiently stopped [23]. We found increased eIF2α phosphorylation and ATF4 expression as an integrated stress response (ISR) under clozapine treatment. The transcription factor ATF4 is a master regulator of the UPR/IRS pathway that mediates cellular responses to ER and mitochondrial stress. It induces the expression of Glut3 [60] and Glut5 as previously described for glioma cells under glucose deprivation [61]. When glucose is scarce, Glut3 is preferentially expressed because it has the highest affinity for glucose among all glucose transporters. Additional upregulation of Glut5 allows for the alternative uptake of fructose [62]. Moreover, ATF4 and glucose deprivation induce HIF1α [63,64], which promotes metabolic switch and upregulates additional survival pathways. In clozapine-treated HL60 cells, we found increased expression of HIF1α and five of its target genes, namely *hsp27* [65], *pdk3* [66], *ATPaseIF1* [67], *slc2a1* (gene coding for Glut1), and *vegf* [68]. VEGF promotes Akt signalling and glucose uptake, while Hsp27 traps and stores misfolded proteins in the ER to avoid their aggregation until refolding or proteolytic degradation [69]. In addition, Hsp27 interacts with HIF1α [70] and activates the anti-apoptotic NF kappa B pathway [71]. The pyruvate dehydrogenase kinase-3 (PDK3) negatively regulates pyruvate dehydrogenase complex activity, and less pyruvate enters the mitochondria [72]. Indeed, pyruvate and citrate levels were decreased in HL60 cell lysates after 24 h.

HIF1α increased Glut expression and pushed cellular metabolism towards glycolysis. Therefore, the glycolytic intermediates were still detectable after 8 h of treatment, while acetyl-CoA and ATP levels were decreased. Under prolonged starvation, glycolysis and PPP are shut down [73], and suppression of pyruvate dehydrogenase complex (PDC) activity by PDK3 together with downregulation of the ETC by IF1 is crucial for glucose conservation and preventing bioenergetic failure. IF1 partially inhibits oxidative phosphorylation at complex V, thereby triggering adaptive metabolic responses leading to drug resistance and preconditioning [74]. This could be exactly the pathway by which clozapine protected neurons of the mouse striatum against kainic acid-induced lesions [75] and serum glucose deprivation [76]. Downregulation of the ETC led to transient accumulation of citrate in the mitochondria, which was transported back by upregulated expression of CiC under clozapine. In the cytoplasm, citrate is the precursor for lipogenesis and is also an allosteric regulator that inhibits glycolytic glucose consumption [77]. Damiano et al. [78] demonstrated that expression of CiC was induced in HepG2 and BRL-3A cells during ER stress. Menga et al. [79] were the first to describe higher CiC expression in clozapine-treated, insulin-secreting Ins-1 cells. In HL60 cells, citrate levels significantly increased upon glucose deprivation, while acetyl-CoA levels decreased within 8 h. However, both returned to baseline levels within 24 h, indicating that the citrate, transported back to the cytoplasm, had refilled the cytosolic pool of acetyl-CoA. Furthermore, citrate is consumed to meet the cellular NADPH demand after the downregulation of the PPP. Although the PPP intermediates strongly declined, the NADP/NADPH ratios did not differ between clozapine-treated and control cells, indicating that the cellular energy status was preserved after 24 h by the citrate shuttle due to the actions of isocitrate dehydrogenase and malic enzyme [73]. Both enzyme levels rise under glucose deprivation. Citrate was metabolised by cytosolic ATP-citrate lyase to acetyl-CoA, which replenished the cellular pool and served for malonyl-CoA synthesis after 24 h. In turn, malonyl-CoA reverses lipolysis [80] to lipogenesis with increased LD formation. LDs are cytoplasmic organelles responsible for storing excess cellular lipids. Their formation is either HIF-dependent or mediated by SREBPs [81]. Interestingly, von Wilmsdorff et al. [82] described sex-dependent activation of SREBPs in the livers of clozapine-feeded rats, provoking either cholesterol synthesis in females or triglyceride and lipid droplet synthesis in males, resembling hepatic steatosis. In addition to lipid storage, LDs contribute to ER homeostasis by the ablation of degraded protein fragments and by ROS scavenging [83]. Previously, Yang et al. [83] could show that LD formation inhibited the induction of autophagy and vice versa. Moreover, autophagy was downregulated in IF1-overexpressing cells [84]. In line with these results, the acridine orange staining of acidic autophagosomes was weak under low clozapine concentrations. Higher concentrations of clozapine (50 µM) provoked autophagy and apoptosis in line with the study of Nury et al. [85]. IF1 mediated downregulation of the ETC by blocking the reversal of ATP hydrolysis of the F0F1-ATPsynthase and preventing dissipation of cellular ATP. This triggers mitochondrial hyperpolarization and the subsequent production of superoxide radicals, which we previously observed in neutrophils from patients under clozapine treatment [42]. LDs and hsp27 reduce cellular ROS [86,87] and thereby improve cellular survival.

HIF1α also dampens oxidative phosphorylation by regulating PDKs [65]. The mitochondrial pyruvate dehydrogenase catalyses the oxidative decarboxylation of pyruvate, and links glycolysis to the Krebs cycle and ATP production. Under clozapine-mediated metabolic reprogramming, elevated PDK3 expression indicates active suppression of the influx of glycolytic metabolites into mitochondria, thereby limiting the citrate-mediated effects on lipid metabolism. Additional hints for in vivo activation of HIF1α under clozapine treatment derive from the bone marrow, where HIF-1α is essential for the mobilization of hematopoietic stem cells [88]. Accordingly, we previously reported significantly increased peripheral CD34+ cell numbers in the first weeks of clozapine treatment [89].

In line with the multiple effects of HIF1α induction, expression of Sirt6, which inhibits the transcriptional activity of HIF1α and ATF4 [90,91], was not induced, while the mitochondrial Sirt3 was upregulated. Sirtuins are metabolic sensors of the cellular energy status. They modulate the activities of key metabolic enzymes via protein deacylation [92]. Sirt3 deacetylates acetyl-CoA synthetase as well as succinate dehydrogenase (SDH), which results in higher acetyl-CoA levels and increased complex II activity [93]. Moreover, Sirt3 counteracts metabolic reprogramming towards glycolysis through Hif1α destabilization [93]. SIRT3 increases resistance against oxidative stress and preserves mitochondrial functions in response to starvation [94]. In contrast, citrate is known to inhibit SDH activity [95], which might be reflected by slightly increased succinate levels in clozapine-treated HL60 cells. Moreover, succinate stabilises HIF-1α under normoxic conditions and induces IL1ß production [96]. Interestingly, increased plasma levels of this cytokine were detected in responders to clozapine treatment, but not in non-responders [97].

Defects in mitochondrial function and complex I in particular have been suggested as a possible cause for the aberrations associated with neuronal diseases [5,98] and the dysfunction of dopaminergic transmission is linked to schizophrenia [99]. Dopamine inhibits the complex I activity, electron transport, and energy supply of the mitochondria [3,4]. Therefore, dopaminergic dysfunction leads to elevated mitochondrial respiration that might contribute to the bioenergetic failure associated with schizophrenia. While some authors described mitochondrial damage due to antipsychotic treatment [100], we here present evidence that clozapine instead induces mitohormesis. ER and mitochondrial stress—like caloric restriction or glucose deprivation—activates the survival pathways UPR and IRS, leading to long-lasting broad metabolic and molecular changes. During starvation, mitochondria fuse into an elongated state [101]. Indeed, Uranova observed significantly larger mitochondria in the cortexes of clozapine-feeded rats already in 1986 [102]. This mitochondrial elongation allows efficient ETC supercomplex formation, which participates in the shift from glycolysis to oxidative phosphorylation [101] and protects from autophagic degradation [103]. This shift from glycolysis to oxidative phosphorylation is reflected by the changed morphology of peripheral neutrophils. When glucose utilization is restricted, peripheral neutrophils show increased immaturity [104]. Consistent with this finding, patients under treatment with clozapine typically have immature neutrophils with reduced nuclear lobularity [105]. In addition to the downregulation of glycolysis, simultaneous expression of IF1 dampens the ETC, resulting in significantly reduced oxygen consumption in HL60 cells. In vitro, this ETC downregulation took 48 h and remained stable for 72 h after clozapine treatment. Thus, IF1 binding to F0F1ATPase contributes to long-term ATP homeostasis without affecting growth but increases cell survival. Moreover, when cells are exposed to nutritional or oxidative stress, IF1 function is not limited to its role as a physiological inhibitor of F0F1 ATPase, but it also constitutes a key component in the metabolic switch of glycolysis to oxidative phosphorylation [106]. Previously, two studies did not find any significant effect of clozapine on the expression and function of mitochondrial complexes [107,108]. Moreover, Amiri et al. [109] revealed that clozapine attenuated mitochondrial dysfunction in an animal model of schizophrenia.

We conclude that clozapine in the low micromolar range significantly changes central carbon metabolism in HL60 cells. On one side, dampened oxidative phosphorylation—which is the most efficient way of energy production—is able to adapt cell metabolism to the decreased intracellular ATP synthesis. On the other side, the upregulation of glucose transporters, together with their VEGF/Akt-mediated localization in the plasma membrane [110], improves glucose/fructose uptake for ATP production. Regarding the inflammatory response at the beginning of clozapine treatment [26], it is conceivable that immune-cell-derived VEGF improves glucose uptake as shown previously at the blood–brain barrier [111].

Given that abnormal cerebral energy metabolism plays an important role in the pathophysiology of schizophrenia, mitohormesis and metabolic reprogramming by clozapine trigger adaptive cytoprotective mechanisms that result in long-lasting broad metabolic and molecular changes. Mitochondrial hormesis, which is defined as an evolutionary-based adaptive response to low-level stress, is emerging as a promising paradigm in the field of neuronal and neurodegenerative diseases. Regarding clozapine`s side effects on blood cells, reduced glucose uptake should be considered as a potential trigger for neutropenia and agranulocytosis at least in patients with reduced Akt activity [7].

### Study Limitations

Low clozapine levels were measurable in the cytosol of HL-60 cells indicating low uptake of clozapine in HL-60 cells [112]. However, the authors used cytosol-containing small intracellular organelles, and the steady recycling of neuroreceptors, including the bound clozapine in the small vesicles, might explain the low level of intracellular clozapine. Although low uptake of clozapine seems unlikely, it cannot be excluded.

mRNA analysis was performed by semiquantitative PCR analysis and gel electrophoresis instead of real-time PCR because we had no access to a real-time cycler.

Akt phosphorylation within the carboxy terminus at Ser473 was not examined.

## 5. Conclusions

Clozapine inhibits signaling via multiple neuroreceptors leading to reduced glucose uptake. Subsequent intracellular glucose deprivation provokes ER and mitochondrial stress. Integrated stress responses activate the Hif1α and ATF4 pathways with upregulated expression of glucose transporters and a metabolic shift from glycolysis to oxidative phosphorylation. Increased mitochondrial formation of superoxide causes an adaptive response that culminates in increased stress resistance and cell survival. Reducing mitochondrial F0F1 ATPase activity by IF1 is linked to reduced oxygen consumption and transient citrate accumulation. Increased expression of the citrate carrier in PBMCs and HL60 cells under clozapine treatment favours the transport of citrate into the cytosol, where it stimulates lipogenesis. HL60 cells and neutrophils from clozapine-treated patients exhibit a significant accumulation of lipid droplets. Consistently, abrogation of the mitochondrial oxidative stress by the green-tea catechin EGCG inhibits prosurvival IF1 induction and increases apoptosis in HL60 cells under cotreatment.

## Figures and Tables

**Figure 1 cells-13-00762-f001:**
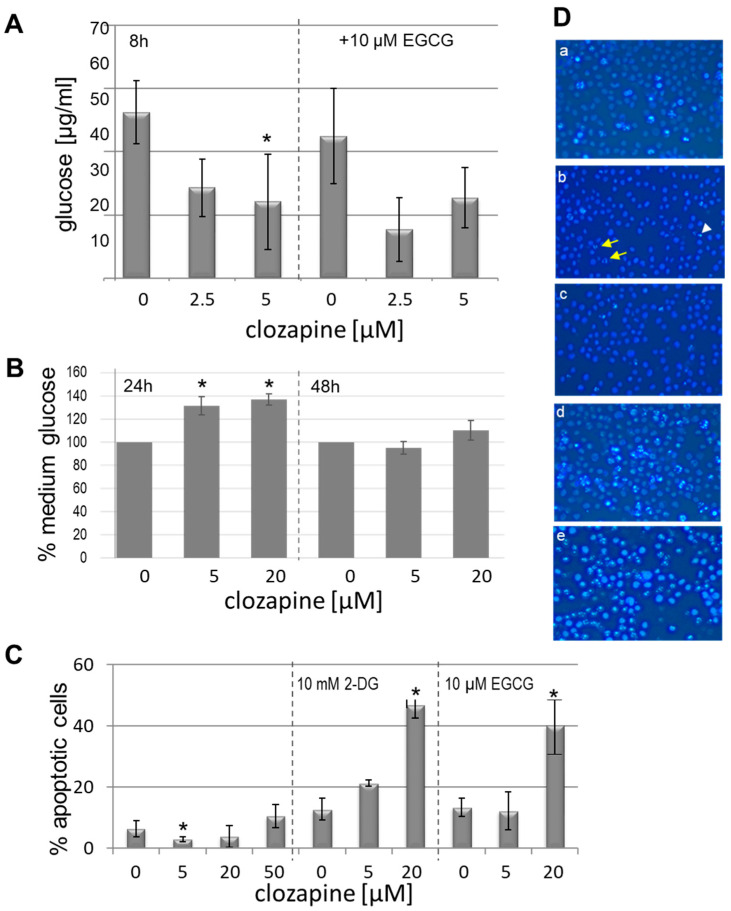
In cultures of clozapine-treated HL60 and DMSO-treated control cells, the intracellular glucose concentration (**A**) was determined after 8 h. Clozapine significantly reduced glucose uptake (*p*  <  0.05, nonparametric Kruskal–Wallis test, * *p*  <  0.05 clozapine (5 μM) vs. DMSO/ethanol control, n  =  3, triplicates each); the influence of EGCG on glucose uptake under clozapine did not reach significance. (**B**) In line with (**A**), the glucose concentration of the medium was significantly increased after 24 h under clozapine treatment. After 48 h, it did not differ between the culture conditions. Bars represent mean ± SD (*p*  <  0.05, nonparametric Kruskal–Wallis test, * *p*  <  0.05 Clozapine (5 μM and 20 µM) vs. DMSO/ethanol control 24 h, n  =  3, triplicates each). (**C**) Clozapine reduced apoptosis in HL60 cells, which was determined after 24 h of incubation (*p*  <  0.05, nonparametric Kruskal–Wallis test, * *p*  <  0.05 Clozapine (5 μM) vs. control; n = 8). EGCG as well as 2-DG significantly increased apoptosis in clozapine-treated HL60 cells. (*p*  <  0.05, Kruskal–Wallis test, * *p*  <  0.05 20 µM clozapine/10 mM 2-DG vs. 2-DG and 20 µM clozapine/10 µM EGCG vs. EGCG, post hoc tests, n  =  3, triplicates each). (**D**) Microscopic detection of Hoechst 33342-stained apoptotic HL60 cells after 24 h of incubation in (**a**) DMSO/ethanol control medium, (**b**) 5 µM clozapine, (**c**) 20 µM clozapine, (**d**) 20 µM clozapine and 10 µM EGCG, and (**e**) 20 µM clozapine and 10 mM 2-DG. Apoptotic cells show pycnotic nuclei with condensed, strongly stained chromatin. The white arrow tip points to a representative apoptotic cell; yellow arrows indicate representative cells in mitosis. On every digital image, at least 200 cells were counted.

**Figure 2 cells-13-00762-f002:**
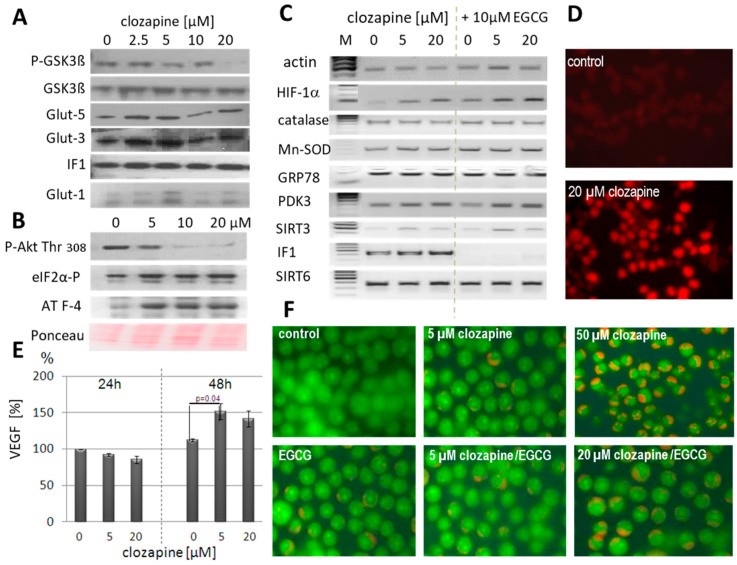
ER stress triggered changes under clozapine. Clozapine-induced changes in gene expression were detected either on the protein level (**A**,**B**,**D**,**E**) or mRNA level (**C**) after 24 h of incubation. Quantitative analyses of the Western blot results are given in Appendix A. (**A**) The ratio of phospho-GSK3ß to GSK3ß significantly decreased under clozapine treatment due to decreased Akt activity and Akt dephosphorylation on threonine 308 (**B**). Protein levels of Glut3, Glut5, ATPaseIF1, and the ER stress proteins phospho-eIF2α, and ATF4 significantly increased under clozapine. Increases in Glut1 failed significance. Expression levels of the DMSO-treated control cells were always set at 100%. (**C**) On the RNA level expression of HIF-1α and its target genes, PDK3, Sirt3, and ATPaseIF1 raised, while the expression of catalase, GRP78 and Sirt6 was unchanged. Coincubation with EGCG impeded ATPaseIF1 expression. Clozapine slightly increased Mn-SOD expression, which was abrogated by EGCG coincubation. Quantitative analysis and further information concerning the gel electrophoresis of the PCR products are given in Appendix A. (**D**) Expression of the HIF-1α target hsp27 strongly increased as part of the stress response (nonparametric Kruskal–Wallis test, *p*  <  0.005 clozapine (20 μM) vs. DMSO/ethanol control, n  =  3). (**E**) VEGF, another HIF-1α target, was determined in the supernatant of HL60 cells by the ELISA technique. After 2 days of incubation with 5 and 20 µM clozapine, VEGF levels were increased (*p* ≤ 0.05 nonparametric Kruskal–Wallis test, *p*  <  0.05 Clozapine (5 μM) vs. DMSO/ethanol control, n  =  3, triplicates each). (**F**) Only a few autophagic lysosomes were stained with acridine orange in clozapine (5 µM) or EGCG-cultured HL60 cells, but clozapine dose-dependently increased the number of acridine orange-stained cells. After incubation with 50 µM of clozapine, nearly 50% of the cells were stained. Quantitative analysis is given in Appendix A.

**Figure 3 cells-13-00762-f003:**
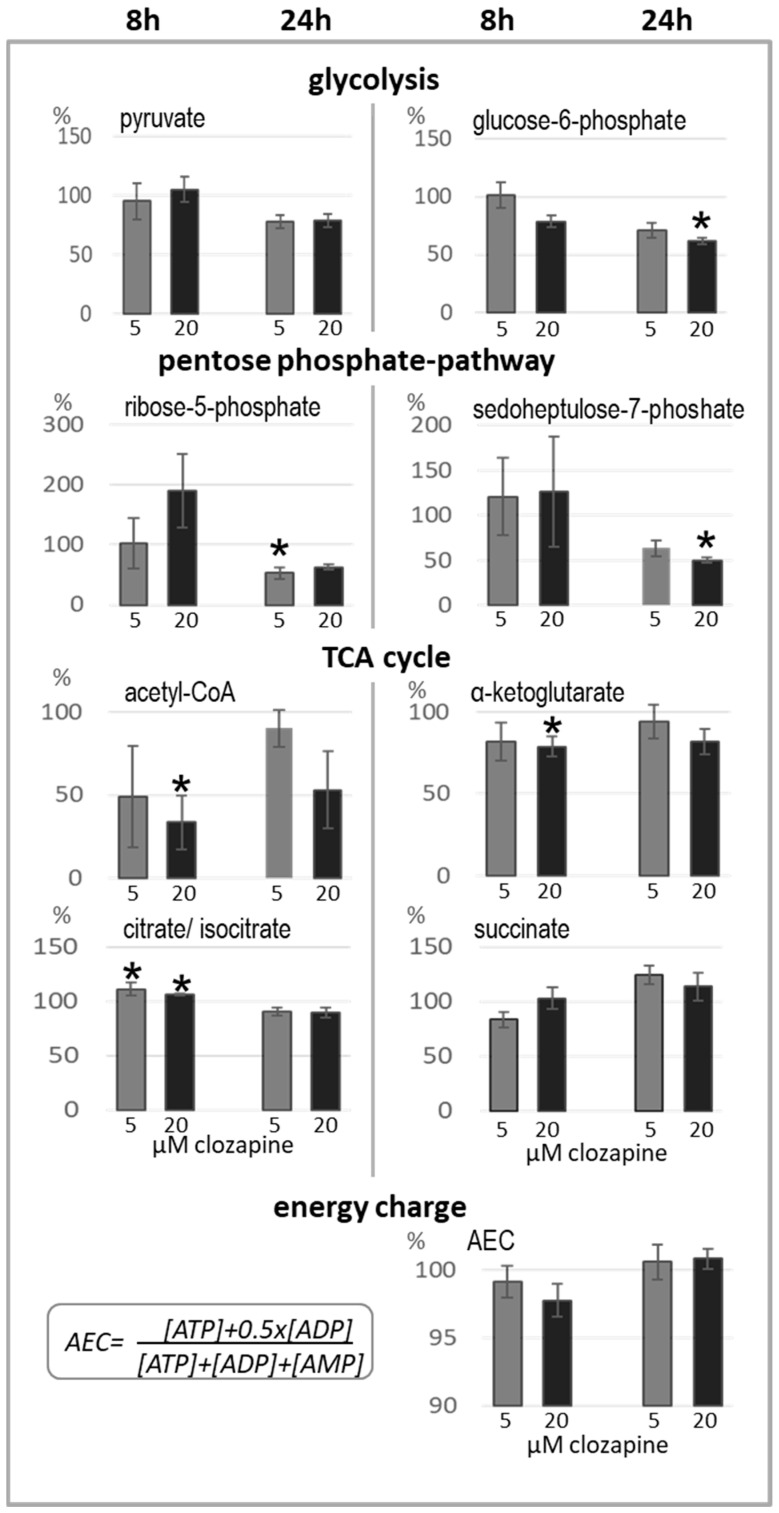
Metabolic changes determined under clozapine treatment for 8 and 24 h. Changes in the main metabolites of glycolysis, PPP and TCA determined by LC–MS/MS measurement under clozapine treatment for 8 and 24 h (n = 3, duplicates each). The energy charge of the cells (AEC) was calculated by the formula presented at the bottom of Figure 3. Bars show % metabolites ± SD in relation to the untreated control cells (100%). Asterisks denote significant differences (* *p* < 0.05) from the values of the control group as calculated by the Wilcoxon test.

**Figure 4 cells-13-00762-f004:**
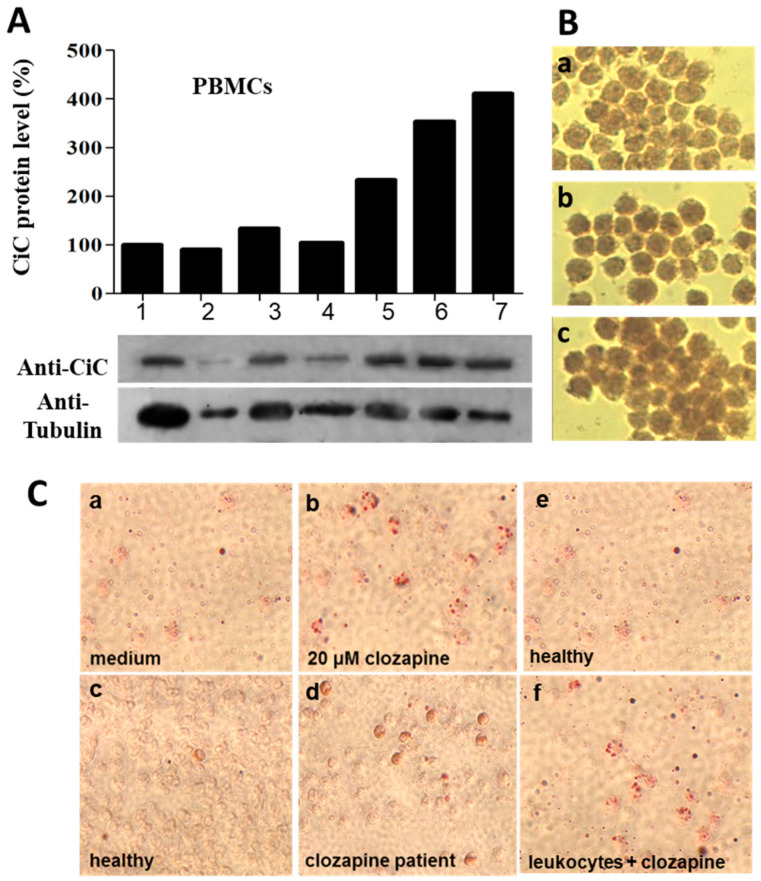
Clozapine increases citrate carrier expression and lipid droplet formation. Western blot analysis of PBMCs (**A**) revealed more citrate carrier (CiC) in PBMCs of three patients under clozapine medication (lanes 5–7) versus individuals without antipsychotic medication (lanes 1–3). Lane 4 shows low citrate carrier levels in a neutropenic patient at the beginning of clozapine treatment. CiC levels were normalised against the housekeeping gene tubulin. (**B**) Immunohistochemical staining of HL60 cells, either untreated (**a**) or treated with 5 µM (**b**) or 20 µM (**c**) clozapine with the CiC antibody (gift from L. Capobianco). For quantitative analysis, the staining intensity for RGB was determined densitometrically. At least 500 cells/treatment were determined. Statistical analysis of the data using Pillai’s trace revealed a significant difference between the groups regarding red, yellow, and blue color components determined for every cell. V = 0.869, F(6, 198) = 23.345, *p* < 0.001. Univariate ANOVAs on the outcome variables revealed significant effects also. Red: F(2, 100) = 146.452, *p* < 0.001; yellow: F(2, 100) = 149.908, *p* < 0.001; blue: F(2, 100) = 85.332, *p* < 0.001. (**C**) Microscopic images of Oil Red-stained LDs in HL60 cells (**a**,**b**) or granulocytes (**c**–**f**) without (**a**,**c**,**e**) or with (**b**,**d**,**f**) clozapine treatment (n = 10 patients). In (**f**), the cells derived from a healthy person (**e**) and were cultured in vitro with 20 µM clozapine for 8 h. At least 500 cells of every condition were counted. The number of cells significantly increased with the clozapine concentration (nonparametric Kruskal–Wallis test, for patients H(2) = 9.000, *p* = 0.003 and HL60 cells H(2) = 7.200, *p* = 0.027). In Appendix A, the results are presented in more detail.

**Figure 5 cells-13-00762-f005:**
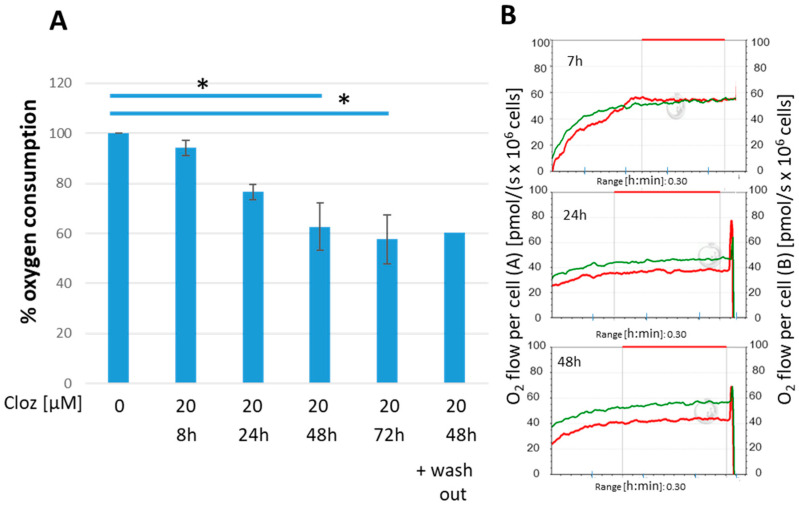
Influence of clozapine on oxygen consumption. High-resolution respirometry revealed strikingly reduced basal oxygen consumption after 48 h of clozapine treatment. (**A**) Bars represent mean ± SD (n = 3, * *p* < 0.05, with respect to DMSO/ethanol control). The oxygen consumption of control cells was set as 100%. After 48 h, the oxygen consumption was significantly reduced (Wilcoxon test * *p* ≤ 0.05). Even after 72 h of washout of clozapine, the effect remained constant (n = 2). (**B**) On the right side of the figure, original tracings of oxygen consumption for untreated HL60 cells (green curves) and cells exposed to 20 µM of clozapine (red curves) for 8/24/48 h are shown. Comparison of the tracings revealed that the oxygen consumption under clozapine (red line) declined with time.

## Data Availability

More data supporting the reported results can be given by the corresponding author.

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
