# Peer review of "Neuroreceptor Inhibition by Clozapine Triggers Mitohormesis and Metabolic Reprogramming in Human Blood Cells"

_cells, 2024, doi:10.3390/cells13090762_

Round 1
Reviewer 1 Report (Previous Reviewer 1)
Comments and Suggestions for Authors
Now this paper is ready to publish in Cells.
Reviewer 2 Report (Previous Reviewer 2)
Comments and Suggestions for Authors
Thanks for corresponding to the comments of the first submission. The reviewer's concerns have been addressed politely by authors and the concerns have been resolved in the current manuscript. If this article is accepted it will be a valuable article, it is speculated that it will have a positive impact on researchers of related fields.
This manuscript is a resubmission of an earlier submission. The following is a list of the peer review reports and author responses from that submission.
Round 1
Reviewer 1 Report
Comments and Suggestions for Authors
Dear Authors,
I have read the manuscript and I send you my comments:
1. In the materials and methods, the authors write that they use concentrations of 2.5-20μM of clozapine, and in the case of apoptosis, a concentration of 50 μM is used. Please explain the discrepancies.
2. What was the reference protein for Western-Blot analysis?
3. Fig. 1 A. Why were only concentrations 2.5 and 5 μM used? Clozapine concentration 20 μM is missing.
4. Fig 1 A and B. Are different EGCG concentrations a mistake?
5. Is there a relationship between lowering glucose levels and lowering the level of apoptotic cells?
6. Fig 2 B. In the case of Western-Blot analysis, concentration 2.5 μM is missing in panel B.
7. In the case of analysis of Gut 1, Gut 3, ATF4 proteins, there is no mass marker.
8. Figure 3. why were only 5 and 20 μM clozapine used?
9. Is there a relationship between lowering glucose-6-phosphate and oxygen consumption?
10. I ask the authors to include information that the membranes were used and marked several times. Please also include information about what buffer or substance was used to strip the membrane to remove antibodies.
Reviewer 2 Report
Comments and Suggestions for Authors
Dr. Fehsel et al have been investigated the effect of clozapine on the mitohormesis and metabolism in HL60 cell and patients. This study is an interesting research topic that could elucidate a molecular mechanism of clozapine treatment in detail and lead to advances in schizophrenia treatment strategies. However, reviewer found several concerns that make the results difficult to interpret, as described below.
In Fig.1A, why was authors investigated only with the low concentrations until 5 μM? In other experiments, higher concentrations from 20 to 50 μM were being considered. Please explain on the reason for setting this study conditions.
In Fig. 1A, the authors described that “Treatment with EGCG provokes a further non-significant decrease of the glucose content…” (at line 201). However, not only the lack of significant difference in results for the glucose levels treated with clozapine and EGCG, but there was an increase at the higher concentration of 5 μM. Why did it increase in 5 μM compared to 2.5 μM? Also, as with other experiments, if authors investigate it at an even higher concentration (10, 20 or 50 μM), it may be considered that the increase even more.
In Fig. 1B, please explain the mechanism by which apoptotic cells increase when treated with high concentrations of clozapine (50 μM).
In Fig. 1B, why was the apoptosis study not conducted at lower concentrations (2.5 μM) like the investigation on the glucose level? Fig 1A and B cannot be compared simply.
In Fig. 1C, reviewer couldn't understand which cells were apoptotic cells from the images.
In Fig. 1C, why were no apoptotic cells observed in Figs c, d and e?
In line 221 to 222, please revise the descriptions to be accurate. Only was considered for low concentration (only 5μM). Also, although it stated 50 μM or more (>50 μM), only 50 μM were conducted in this study.
In line 228 to 229, it was described that Akt phosphorylation has decreased, but since Figure 2 only showed the bands and no values, it could not be judged. Please show all the experimental results in Figure 2 in values, not in just images or some conditions only.
On ER stress study. Including Supplemental Fig. 1, only data for 2.5 and 5 are graphed. Therefore, reviewer was unable to read about the decrease in phosphorylation of threonine 308 and GSK3b, the increase in eIF2α and the associated upregulation of stress proteins, and the increase in ATF4 and the associated upregulation of glucose transporters and hsp27. Please quantify or graph all the results, not just some of the data. It may be suspected from readers that this is a deliberate selection, and it does not resolve the issue.
In Fig 2A, B, and C, Reviewer couldn't understand the results because it couldn't judge the increase or decrease from the bands image alone. Please quantify the blotting data.
In the VEGF study (in Fig. 2E), other experiments have been conducted up to 24 hrs, while why 48 hours was investigated? Also, why didn't VEGF increase after 24 hr?
In Fig 2E, an increase was also observed in the clozapine untreated group (0) from 24 to 48 hours. Why was VEGF increased over time in the untreated group? This result may suggest that there are an effect of the increase with time in the treatment group as well.
In line 265, authors described that VEGF was increased only in 5 μM. So why didn't it increase significantly at 20? AIF4 was increased sufficiently at 20 μM. Please explain this discrepancy.
In Fig. 1F, it was not possible to understand the results based only on the image of acridine orange. Please quantify it so that readers can make quantitative decisions.
In Fig. 2C, reviewer could not understand the results from only the bands images. Please represent as the values.
In Fig. 3, Why did authors conduct the investigation for 8 hours in the metabolic study? The time course was not investigated in other experiments.
In line 300, authors described that “All TCA-metabolite levels normalized within 24h …”. After 24 hours, why did the acetyl CoA decrease at 20 μM compared to 5 μM?
In Fig. 4B, reviewer could not understand the difference between each image. Please represent quantitatively and add the descriptions based on the result.
In Fig. 4C, regarding the LD formation, only one image is shown for each condition, and it is not possible to make a judgment based only on this partial images. Please quantify in the entire tissue and describe and discuss it based on objective results.
In Fig. 5, regarding the Oxygen consumption, why was the investigation conducted for 7 hours? In other experiments it was not considered for this time course.
